# Mechanism of Accelerant on Disperse Dyeing for PET Fiber in the Silicone Solvent Dyeing System

**DOI:** 10.3390/polym11030520

**Published:** 2019-03-19

**Authors:** Jiping Wang, Wenqing Cheng, Yuanyuan Gao, Lei Zhu, Liujun Pei

**Affiliations:** 1Engineering Research Center for Eco-Dyeing and Finishing of Textiles, Zhejiang Sci-Tech University, Hangzhou 310018, China; jpwang@zstu.edu.cn (J.W.); wqcheng0725@outlook.com (W.C.); aloysyy@163.com (Y.G.); lzhuzj@163.com (L.Z.); 2Schhool of Fashion Engineering, Shanghai University of Engineering Science, Shanghai 201620, China; 3National Base for International Science & Technology Cooperation in Textiles and Consumer-Goods Chemistry, Zhejiang Sci-Tech University, Hangzhou 310018, China

**Keywords:** PET fiber, disperse dye, silicone solvent, accelerant, solubility, swelling

## Abstract

Disperse dyeing for polyethylene terephthalate (PET) fiber in different non-aqueous solvent dyeing systems have been extensively studied over the past decades. In the present work, disperse dyeing for PET was investigated in a silicone solvent dyeing system. The influence of accelerant on the fiber swelling, uptake of dye, K/S value of dyed fiber, and dye solubility in the silicone solvent were systematically investigated. Compared with no accelerant, the final uptake of the disperse dye (C. I. Disperse Blue 367) could increase to 81% with 20% accelerant in the silicone solvent dyeing system, and the K/S value of dyed fiber was also higher (3.3 for no accelerant vs. 13.2 for accelerant). The influence of accelerant on the performance of disperse dyeing was also studied. Firstly, the solubility of the disperse dye in the silicone solvent can be decreased by the accelerant. Moreover, the solubility of the disperse dye is inversely proportional to the K/S value and the uptake of the dye. In addition, although the silicone solvent can diffuse to the inner fiber and has a partial swelling in the PET fiber, the swelling of PET can be improved by the accelerant. Furthermore, the swelling of fiber can reach equilibrium when the amount of accelerant was 15% (the weight of fiber). Therefore, this eco-friendly dyeing technology has considerable potential for application to a broad array of chemical fibers.

## 1. Introduction

For the traditional dyeing of PET (polyethylene terephthalate), commercial disperse dyes are generally used for dyeing. However, a larger amount of dispersant is used in the water base because the dispersion of disperse dye is poor in water [1,2]. As a result, disperse dyeing in aqueous solutions brings about problems such as the assumption of a large amount of water and hazardous industrial effluents, leading to large energy and environmental challenges [3,4,5,6,7]. Therefore, much research is focused on non-aqueous systems for dyeing PET with disperse dye, such as the liquid paraffin dyeing system [8,9], supercritical carbon dioxide dyeing system [10,11,12], and organic solvent dyeing system [13,14,15].

Among the previous non-aqueous dyeing systems, supercritical carbon dioxide dyeing can achieve satisfactory dyeing performance due to its excellent properties, like viscosity, density, diffusivity, and dielectricity [11]. Unfortunately, industrialization is extremely difficult due to various factors such as the high cost of dyeing equipment and potential dangers of operation (e.g., the operating pressure of the equipment is very high) [16,17,18,19]. Another approach is to dye polyester textile in the solvent using an organic solvent as the dyeing medium [14]. However, this dyeing technology was often carried out by using hydrocarbon solvents which are not environment-friendly, such as hexane, cyclohexane, and n-heptane, as the continuous phase medium [15].

In recent years, some researchers have begun to choose new dyeing media to achieve a suitable eco-friendly dyeing system [20]. In our previous investigations [21,22], D5 (decamethylcyclopentasiloxane) was chosen as a dyeing medium to prepare a silicone solvent dyeing system for dyeing PET with disperse dye. The chemical structure of D5 is shown in Figure 1 [23,24,25]. In the D5 dyeing system, PET can be dyed with pure disperse dye without adding any dispersant. Furthermore, the dyeing medium can be recycled after dyeing. The dyeing performance is equivalent to that of traditional high temperature and high-pressure dyeing.

In our previous investigations, Wu et al. [22] reported that the strength of D5-dyed PET is similar to the PET after dyeing in conventional water baths, which indicates that D5 dyeing has little effect on the strength of polyester fabrics. The soaping and rubbing fastness are about 4~5 grade [22]. Li et al. [21] reported that the uptake of disperse dye in D5 solvent is much lower than that in the traditional water bath, though adding a small amount of accelerant can significantly increase the uptake of disperse dye on polyester fiber. However, there is no corresponding reasonable explanation in the previous report.

To the best of our knowledge, a system for dyeing PET fiber with disperse dye in D5 solvent has rarely been reported. Therefore, in this work, we first studied the impact of a small amount of accelerant on fiber swelling. The effect of accelerant content on dye uptake and solubility of dye in the D5 solvent were further investigated.

## 2. Experimental Procedure

### 2.1. Materials

The polyester fiber (3.5D × 65 mm) used for the dyeing performance was collected from Wanziqianhong Co., Ltd., Haining, China. C. I. Disperse Red 177, C. I. Disperse Blue 367, and C. I. Disperse Orange 30 (filter cake, no dispersant) were received from Zhejiang Longsheng Dye Chemical Co., Ltd., Shangyu, China. The molecular structures of the dyes are shown in Figure 2. An accelerant was purchased from GCH Technology Co., Ltd., Shenzhen, China. The fluorescent labeling dye Rhodamine B (accelerant-soluble, red), Coumarin 6 (D5-soluble, green; HPLC grade), and DMSO (Dimethyl sulfoxide, analytical reagent) were purchased from Aladdin (Shanghai, China). Ammonium sulfate (analytical reagent) was obtained from Hangzhou Gaojing Chemical Reagent Co., Ltd., Hangzhou, China. D5 (purity > 96%) was purchased from GE Toshiba Silicone Ltd., Jiande, China.

### 2.2. Dyeing Method

Dyeing was carried out on a DYE-24 dyeing machine (ShangHai Chain-Lih Automation Equipment Co., Ltd., Shanghai, China). Firstly, 5 g polyester fiber was dyed with 0.5% (the weight amount of fiber) disperse dye and a different amount of accelerant (0%, 5%, 10%, 15%, 20%, 25%; the weight amount of fiber) at a liquor ratio of 20:1 in D5 media. Dyeing was started at room temperature with disperse dye and accelerant, and then the temperature was raised to 140 °C for 60 min. After dyeing, the dyed fiber was washed twice using D5 at a liquor ratio of 20:1 at 80 °C for 15 min.

### 2.3. Dyeing Evaluations

The color depth of the dyed fiber in percentage was calculated first by determining the reflectance R of the dyed fiber at the wavelength of minimum reflectance (maximum absorbance) on a Datacolor SF600X spectrophotometer (Datacolor, Lawrenceville, NJ, USA). The color yield (*K*/*S*) value was then calculated using the Kubelka–Munk equation (Equation (1)) [3]:(1)$$\frac{K}{S}=\frac{1-R^{2}}{2R}$$

The dye uptake rate is the dye on fiber to the percentage of dye which is added during dyeing. The dye in the raffinate was extracted with DMSO. The dye fixation rate was evaluated using Equation (2):(2)$$E=(1-\frac{C_{1}V_{1}}{C_{0}V_{0}})\times 100\%$$
where *E* refers to the dye uptake rate; *C*_0_ and *C*_1_ refer to the concentration of initial dye (g/L) and the concentration of all the raffinate, respectively; and *V*_0_ and *V*_1_ refer to the volume (mL) of the initial dye bath and the volume (mL) of all the raffinate, respectively.

### 2.4. The Solubility of Dye

Excess dyes and accelerants were added in the D5 media at room temperature, and then the temperature was raised to 140 °C for 60 min. The dyes that were dissolved in D5 were extracted with DMSO. The solubility of dye was calculated by using Equation (3):(3)$$R=\frac{M}{V}$$
where *R* refers to the dye solubility (g/L), *M* refers to the weight of dissolved dye (g), and *V* refers to the volume of D5 (L).

### 2.5. Fiber Swelling

Firstly, the degree of fiber swelling needed to be confirmed with a different accelerant content. The method was as follows:

Before dyeing, 50 fibers were selected randomly, and all the fibers were photographed with confocal laser scanning microscopy (CLSM; NikonC2, Nikon, Tokyo, Japan). The magnification was 400 times. Five positions for each fiber were picked to measure the fiber diameter, and then the average value and standard deviation were calculated. Among them, ten of the largest diameter fibers and ten of the smallest diameter fibers were removed. Moreover, the remaining mean and standard deviations were calculated.

After dyeing, fibers were centrifuged at 4000 r/min for 10 min. Fifty fibers were selected randomly. The processing method was the same as above. Fiber swelling was calculated using Equation (4):(4)$$S=\frac{L_{1}-L_{0}}{L_{1}}\times 100\%$$
where *S* refers to the fiber swelling ratio, and *L*_0_ and *L*_1_ refer to the average fiber diameter before and after dyeing, respectively.

If the PET fiber could be swelled by D5 media, the dyeing media must enter the interior of fiber. As a result, the silicon content that diffuses into the fiber cross-section before and after dyeing was measured with X-ray photoelectron spectroscopy (XPS; K-Alpha Thermo Fisher Scientific, Waltham, MA, USA). Furthermore, one accelerant-soluble fluorescent dye was selected to dye the D5 bath (15% accelerant, the weight of fiber) with a D5-soluble fluorescent dye. After dyeing, the fiber cross-sections needed to be photographed under CLSM in the G-2A and the FITC (Fluorescein Isothiocyanate) fluorescence channel.

## 3. Results and Discussion

### 3.1. K/S Value and Dye Uptake of Disperse Dye in the D5 Solvent Dyeing System

Figure 3 shows the relationship between the amount of accelerant and the K/S value of the fiber. The K/S value of dyed fiber increased with the increase of accelerant. For example, the *K*/*S* value of C. I. Disperse Blue 367 was only 3.3 without accelerant. However, it increased to 13.2 when the amount of accelerant was 20% (the weight of dyed fiber). The same situation occurred in the other two dyes (Figure 3a). Obviously, the addition of accelerant can effectively improve the color depth of PET, which indicated that the dye uptake is different with different amounts of accelerant.

As shown in Figure 3b, the dye uptake of dye also showed changes with the addition of accelerant. For example, the dye uptake of C. I. Disperse Blue 367 was only 66.5% without accelerant, but it increased with the increase of accelerant. When the amount of accelerant was 20%, the dye uptake of C. I. Disperse Blue 367 reached 81%. The same situation occurred in the other two dyes. As a result, the accelerant is an important factor affecting the silicone solvent dyeing of PET fiber. Furthermore, the addition of accelerant can effectively improve the dye uptake.

### 3.2. The Solubility of Disperse Dye in D5 Solvent

Figure 4 shows that a low amount of accelerant can significantly improve the dye uptake and the *K*/*S* value. In traditional solvent dyeing, the disperse dye has a higher solubility in the solvent, resulting in a lower partition coefficient of the dye [26,27]. Therefore, reducing the solubility of the dye in the solvent can effectively improve the dye uptake. The siloxane solvent dyeing system is similar to the traditional solvent dyeing system. An accelerant can greatly improve the dye uptake, so it is necessary to study the effect of accelerant on the solubility of disperse dye in the D5 solvent.

The effects of the different amount of accelerant on the solubility of dye are clearly shown in Figure 4. The solubility of the dye decreased with the increase of accelerant. For example, the solubility of C. I. Disperse Orange 30 was 1.02 g/L without using any accelerant in the D5 solvent. However, the solubility of dye could decrease to 0.44 g/L when the amount of accelerant was 15% (the weight of dyed fiber). Similarly, the solubility of C. I. Disperse Red 177 and C. I. Disperse Blue 367 in D5 solvent was 0.087 g/L and 0.68 g/L, respectively. When the amount of accelerant was 20%, the solubility of C. I. Disperse Red 177 and C. I. Disperse Blue 367 decreased to 0.065 g/L and 0.31 g/L, respectively, indicating that dye solubility of disperse dye in D5 could be reduced by a small amount of accelerant. The optimum amount of accelerant was different when the solubility of different disperse dyes reached a balance.

Dye solubility was linked to dye uptake and the *K*/*S* value. Therefore, the effect of dye solubility on the K/S and dye uptake was investigated using solubility as the abscissa.

As shown in Figure 5, in the D5 solvent dyeing system, both the *K*/*S* value and the uptake of dye decreased with the solubility of disperse dye. In general, the uptake of dye can influence the *K*/*S* value. The lower the uptake of dye, the smaller the *K*/*S* value [28,29]. Because the dye solubility is inversely proportional to the uptake of dye, the dye tends to diffuse into the PET fiber by reducing the solubility of dye in the dye bath. As a result, more disperse dye would diffuse into the inner regions of the PET fiber, and a greater color depth of fiber could be achieved in the D5 solvent dyeing system. In addition, accelerant affected dyeing by decreasing dye solubility.

### 3.3. The Effect of Accelerant Content on Fiber Swelling in the D5 Solvent

Generally, PET fiber can achieve a well-swelling in the traditional water base because the water molecules can facilitate the diffusion of dyes into the interior of the fiber [30,31]. As shown in Figure 6, different amounts of accelerant have different effects on fiber swelling. The swelling rate of PET fiber increased with the increase of accelerant in the D5 solvent dyeing system. When the content of accelerant was 15%, the swelling of PET fiber reached equilibrium (15%). For the swelling of PET fiber, the optimal amount of accelerant was about 15%. However, the optimal amount of accelerant might differ. As a result, the PET fiber can achieve a well-swelling by adding accelerant in the D5 solvent dyeing system. The reason may be that the chain mobility of PET was enhanced by the accelerant at a higher temperature (dyeing temperature was about 140 °C), which can create larger free volumes, and thus more disperse dyes can diffuse into the interior of the PET fiber [32,33]. Therefore, accelerant affects dyeing by improving fiber swelling.

### 3.4. The Distribution of D5 in the Interior of PET Fibers

PET fiber still swell without adding any accelerant in the D5 solvent, although the fiber swelling ratio was only 3.4%. This meant that D5 also has a partial swelling effect on the PET fiber. As a result, D5 might diffuse to the interior of the fiber.

As shown in Table 1 and Figure 7, Si content was 0.83% in the fiber cross-section before dyeing; however, it increased to 1.52% after dyeing, indicating that the silicon content increased by 83.13% after dyeing. As a result, the D5 solvent can enter the interior of the PET fiber. Furthermore, the silicon content remained the same whether the accelerant was added during the dyeing process, indicating that the dyeing accelerant had no effect on the distribution of D5 in the interior of the fiber.

In addition, the distribution of D5 was investigated by observing the distribution of fluorescent dye in the cross-section of PET fibers. As shown in Figure 8, the accelerant-soluble fluorescent dye was observed under the G-2A fluorescence channel, and D5-soluble fluorescent dye was observed under the FITC fluorescence. Moreover, both fluorescent dyes were observed under the merged channel. As a result, not only can the accelerant diffuse into the inner of PET fiber, the D5 solvent can also diffuse into the interior of PET fiber. From the above results, it can be concluded that the D5 solvent can diffuse into the interior of PET fibers, and the PET fiber can be swelled by the D5 solvent in the D5 dyeing system.

## 4. Conclusions

In the silicone solvent dyeing system, a small amount of accelerant can effectively increase the dye uptake of dye and the color depth of PET fiber. Firstly, the solubility of disperse dye can be decreased by the accelerant in the D5 solvent. Moreover, the solubility of disperse dye is inversely proportional to the *K*/*S* value and the dye uptake, resulting in the tendency of the disperse dye to diffuse into the PET fiber by reducing the solubility of the dye. For the swelling of PET fibers, D5 can diffuse to the interior of the fiber and has a partial swelling in the PET fiber. However, the swelling of PET was mainly improved by the accelerant. When the accelerant content is 15% (the weight of fiber), the swelling of fiber reached equilibrium. Moreover, the optimum amount of accelerant for different disperse dyes is different because the solubility of the dyes is different in the D5 solvent. The favorable results implied that the accelerator can effectively improve the dye uptake of disperse dye. Under the trend of green production, eco-friendly dyeing technology is highly sought. The continuous research and improvement of silicone solvent dyeing technology can lead the industrialization of eco-friendly dyeing technology.

## Figures and Tables

**Figure 1 polymers-11-00520-f001:**
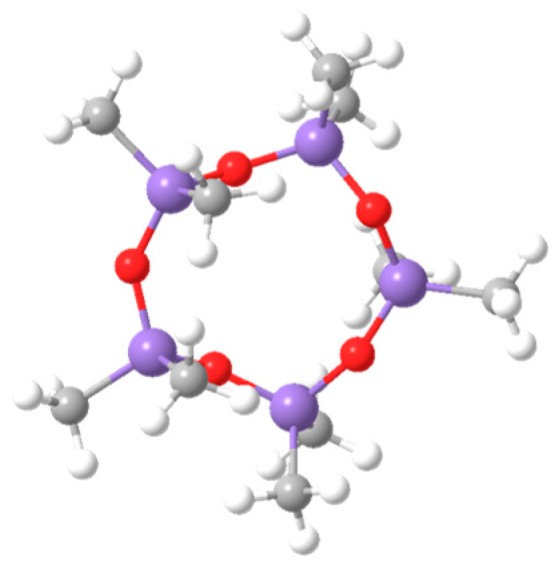
The chemical structure of decamethylcyclopentasiloxane (D5).

**Figure 2 polymers-11-00520-f002:**
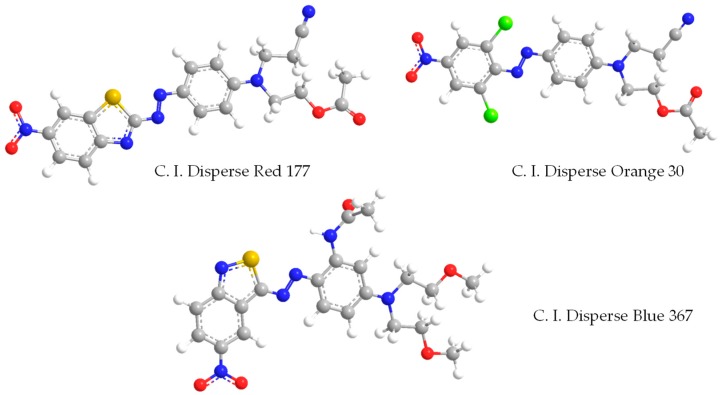
Molecular Structures of disperse dyes.

**Figure 3 polymers-11-00520-f003:**
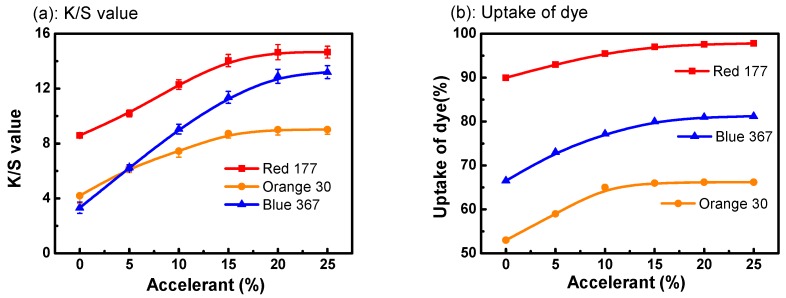
Disperse dying with different amount of accelerant in D5 solvent dyeing system: (**a**) the K/S value of dyed fiber; (**b**) the uptake of dye.

**Figure 4 polymers-11-00520-f004:**
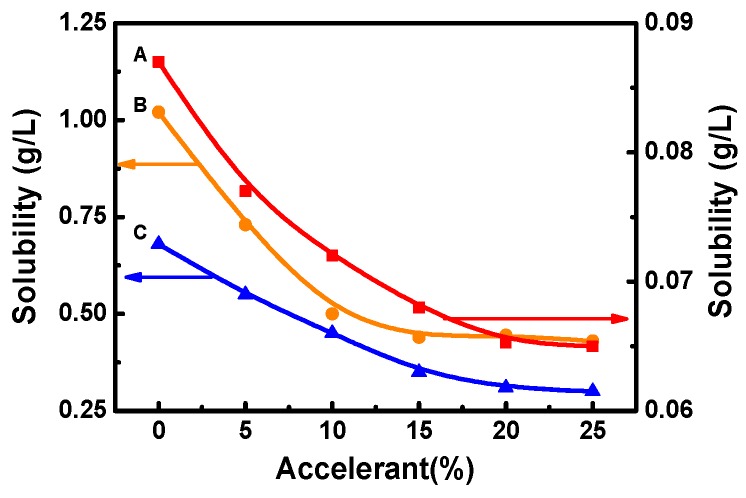
The solubility of three disperse dyes in the D5 solvent (A: C. I. Disperse Red 177, B: C. I. Disperse Orange 30, and C: C. I. Disperse Blue 367).

**Figure 5 polymers-11-00520-f005:**
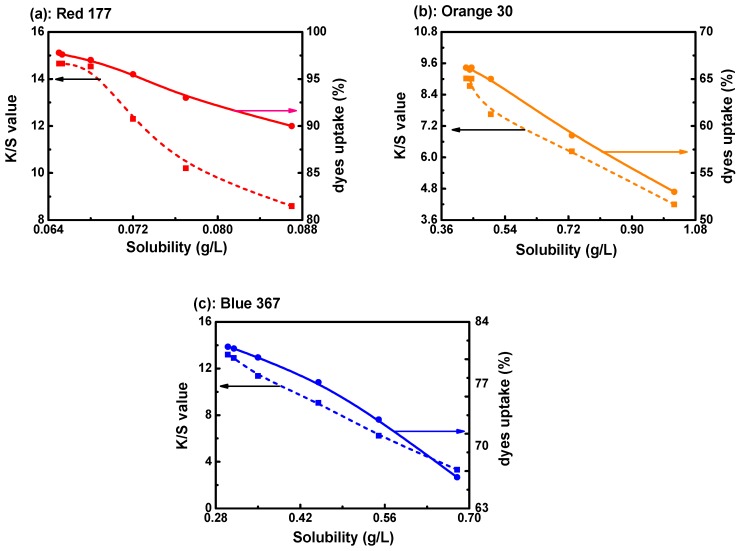
The relationship between dye solubility and K/S value and dye uptake: (**a**) C. I. Disperse Red 177, (**b**) C. I. Disperse Orange 30, and (**c**) C. I. Disperse Blue 367.

**Figure 6 polymers-11-00520-f006:**
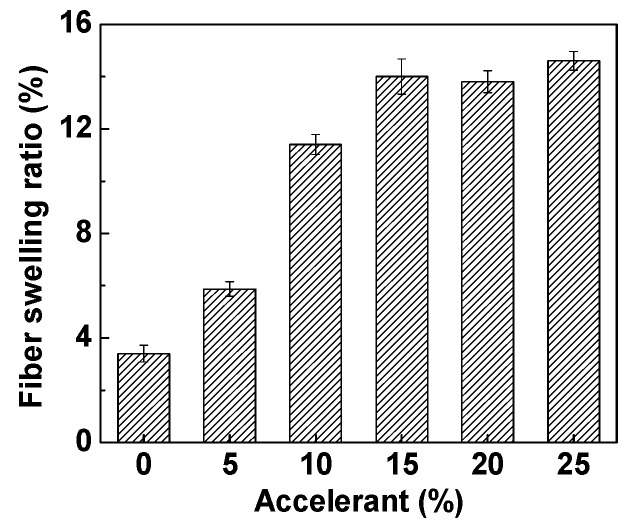
The swelling of PET fiber with different contents of accelerant in the D5 solvent.

**Figure 7 polymers-11-00520-f007:**
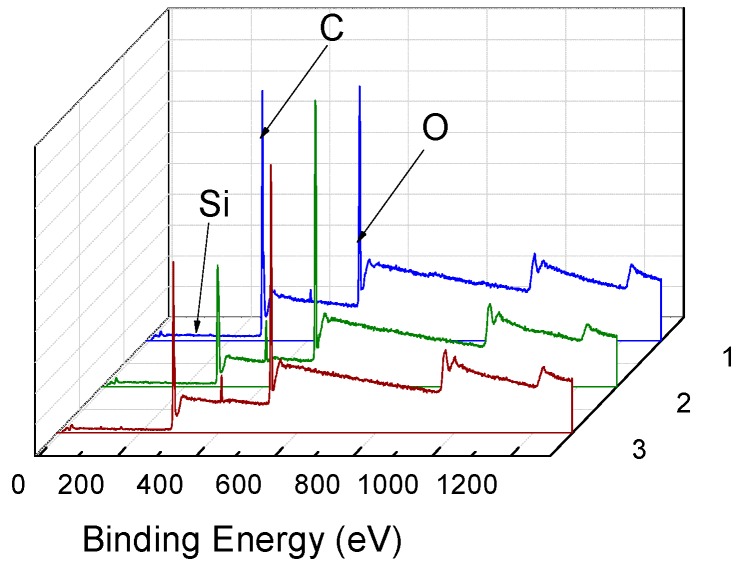
Wide scan XPS of PET fiber before and after dyeing (**1**: B\before dyeing. **2**: after dyeing without adding accelerant, **3**: after dyeing with accelerant).

**Figure 8 polymers-11-00520-f008:**
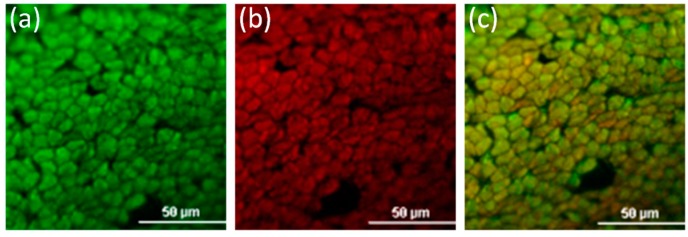
The distribution of accelerant and D5 in PET fiber cross-sections: (**a**) FITC channel in D5, (**b**) G-2A channel in accelerant, and (**c**) merged channel.

**Table 1 polymers-11-00520-t001:** Si content in the fiber cross-section after dyeing.

Element	Atomic %		
	Before Dyeing	After Dyeing without Adding Accelerant	After Dyeing with 20% Accelerant
C	63.37	63.29	63.23
O	35.80	35.19	35.25
Si	0.83	1.52	1.52

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
