# Peer review of "Mechanism of Accelerant on Disperse Dyeing for PET Fiber in the Silicone Solvent Dyeing System"

_polymers, 2019, doi:10.3390/polym11030520_

Reviewer 1 Report

The technical content is sound good and minor revision is required:

(i) The solvent recovery rate should be reported.

(ii) The mechanical properties, e.g. tensile strength, of the D5-dyed PET should be reported.

(iii) Colour fastness properties should be reported.

Author Response

Response to Reviewer 1

Comments Point 1: The solvent recovery rate should be reported.

Response 1: We appreciate the reviewer’s comment. This siloxane medium was reused by extraction. The recovery rate of dyeing media can reach to 99%. In the future, more research will be done on recycling.

Point 2: The mechanical properties, e.g. tensile strength, of the D5-dyed PET should be reported.

Response 2: We appreciate the reviewer’s comment. The mechanical properties of the D5-dyed PET had been investigated in our previous investigations [22]. Compared with the PET after dyeing in conventional water bath, the strength of D5-dyed PET change a little, which indicates that D5 dyeing has little effect on the strength of polyester fabric [22]. Change: ...In our previous investigations, Wu et al. [22] reported that the strength of D5-dyed PET is similar with the PET after dyeing in conventional water bath, which indicates that D5 dyeing has little effect on the strength of polyester fabric... 

Point 3: Colour fastness properties should be reported.

Response 3: We appreciate the reviewer’s comment. Colour fastness properties had been investigated in our previous investigations [22]. The soaping and rubbing fastness are about 4~5 grade [22]. Change: ...In our previous investigations, Wu et al. [22] reported that the strength of D5-dyed PET is similar with the PET after dyeing in conventional water bath, which indicates that D5 dyeing has little effect on the strength of polyester fabric. The soaping and rubbing fastness are about 4~5 grade [22]. Li et al. [21] reported...

Reviewer 2 Report

I think the manuscript would be worth of publishing in the Journal providing following remarks/suggestions will be accounted for. I hope that they may help improving the manuscript appropriately. Alternatively, the authors should give reasonable comments and answers.

1.It is not necessary to include all of your data in a single article. Also use statistics appropriately, and be clear on the need to justify the sophistication of analysis in the context of the validity of underlying data.
2.I congratulate you on producing a paper that, , is for the most part very clear and understandable. I very much appreciate the effort that this requires.
3.Please comment on factors affecting the silicone solvent dyeing of PET fiber4.In conclusions, it would be helpful if authors exhibit outlook towards eco-friendly dyeing technology, point to any current research efforts in this direction.

Author Response

Response to Reviewer 2 Comments

Point 1: It is not necessary to include all of your data in a single article. Also use statistics appropriately, and be clear on the need to justify the sophistication of analysis in the context of the validity of underlying data.

Response 1: We appreciate the reviewer’s comment. The data is expressed in the form of a figure to more intuitively observe the laws in the data. Due to fiber swelling and dye solubility are both effected by the accelerant, some experiments need to be designed to verify the primary and secondary relationships among them in the future.

Point 2: I congratulate you on producing a paper that, , is for the most part very clear and understandable. I very much appreciate the effort that this requires.

Response 2: Thank you very much for your affirmation of our work.

Point 3: Please comment on factors affecting the silicone solvent dyeing of PET fiber

Response 3: We appreciate the reviewer’s comment. In addition to the dyeing time and temperature, the swelling of fiber, the solubility of dye, and the dosage of accelerant are important factors. But experiments have found that fiber swelling and dye solubility are both effected by the accelerant.  

Change: ...As a result, accelerant is important factor which dose affect the silicone solvent dyeing of PET fiber. What’s more, the addition of accelerant can effectively improve the dye uptake.

...As a result, more of disperse dyes would diffuse into the inner of PET fiber, and a deeper color depth of fiber could be achieved in the D5 solvent dyeing system. In addition, accelerant affect dyeing by decreasing the solubility of dye.

...disperse dyes can diffuse into the inner of PET fiber [32, 33]. Therefore, accelerant affect dyeing by improving the swelling of fiber.

Point 4: In conclusions, it would be helpful if authors exhibit outlook towards eco-friendly dyeing technology, point to any current research efforts in this direction.

 Response 4: We appreciate the reviewer’s comment. Under the trend of green production, eco-friendly dyeing technology is highly concerned. With the continuous research and improvement of silicone solvent dyeing technology, it will lead the industrialization of eco-friendly dyeing technology.

Change: ...the dye uptake of disperse dye.Under the trend of green production, eco-friendly dyeing technology is highly concerned. With the continuous research and improvement of silicone solvent dyeing technology, it will lead the industrialization of eco-friendly dyeing technology.
